# Taking the chance!–Interindividual differences in rule-breaking

**Leidy Cubillos-Pinilla** [1]*, **Franziska Emmerling**[2]

**1** Neurophysiology Leadership Laboratory, Technical University München–School of Management, Chair of Research and Science Management, Munich, Germany, **2** Marie Skłodowska-Curie Actions Post-Doctoral Fellow at the Technical University München–School of Management, Chair of Research and Science Management, Head of Neurophysiology Leadership Laboratory, Munich, Germany

\* leidy.cubillos-pinilla@tum.de

**Data Availability Statement:** All relevant data are within the manuscript and its Supporting Information files.

**Funding:** FE (Dr. Franziska Emmerling) MARIE SKŁODOWSKA-CURIE ACTIONS Individual

## Abstract

While some individuals tend to follow norms, others, in the face of tempting but forbidden options, tend to commit rule-breaking when this action is beneficial for themselves. Previous studies have neglected such interindividual differences in rule-breaking. The present study fills this gap by investigating cognitive characteristics of individuals who commit spontaneous deliberative rule-breaking (rule-breakers) versus rule-followers. We developed a computerised task, in which 133 participants were incentivised to sometimes violate set rules which would–if followed–lead to a loss. While 52% of participants tended to break rules to obtain a benefit, 48% tended to follow rules even if this behaviour led to loss. Although rule-breakers experienced significantly more cognitive conflict (measured via response times and mouse movement trajectories) than rule-followers, they also obtained higher payoffs. In rule-breakers, cognitive conflict was more pronounced when violating the rules than when following them, and mainly during action planning. This conflict increased with frequent, recurrent, and early rule-breaking. Our results were in line with the Decision-Implementation-Mandatory switch-Inhibition model and thus extend the application of this model to the interindividual differences in rule-breaking. Furthermore, personality traits such as extroversion, disagreeableness, risk propensity, high impulsiveness seem to play a role in the appreciation of behaviours and cognitive characteristics of rule-followers and rule-breakers. This study opens the path towards the understanding of the cognitive characteristics of the interindividual differences in responses towards rules, and especially in spontaneous deliberative rule-breaking.

## 1. Introduction

Humans tend to follow norms because this action is reinforced by peers, superiors, and society [1–6]. Even seemingly simple behaviours such as verbal communication are grounded in surprisingly complex rules with respect to grammar and pragmatics [7, 8]. Most of the time, human agents effortlessly follow such regulations as these norms define behaviours that are allowed and expected in specific social situations [9–11]. Rule-following can, furthermore, be

Fellowships (IF) grant, Call: H2020-MSCA-IF-2017
Link: https://ec.europa.eu/info/funding-tenders/
opportunities/portal/screen/opportunities/topic-
details/msca-if-2017 The funders had no role in
study design, data collection and analysis, decision
to publish, or preparation of the manuscript.

**Competing interests:** The authors have declared
that no competing interests exist.

favourable for individual agents because rule-followers are perceived as more reliable social partners [12–14]. The described advantages seem to solidify rule-following behaviour as a default mode for cultural evolution [15, 16].

Despite rule-following advantages, humans also show a tendency to break rules when established conventions thwart their attempts to succeed. Although rule-following is the dominant behavioural action plan, rules are broken if the value of the expected outcome following this action is sufficiently large (e.g., increased reward, increased social desirability, expedited task completion; [2, 16–18]). Interestingly, some individuals tend to follow rules regardless of the consequences [19–21], while others tend to make an effort to break them specifically to obtain benefits [18, 22–25]. These interindividual differences seem to be more palpable in individualistic cultures in which individuals are more prompted to commit rule-breaking in comparison to collectivistic cultures [26]. This study investigates cognitive and personality characteristics of these interindividual differences in spontaneous deliberative rule-breaking (rule-followers versus rule-breakers) in an individualistic culture and results cannot be generalised to collectivistic settings. Notably, this research focuses on general norms rather than on social, legal or moral norms. That is, we investigate individuals' reaction towards the connotation of framing a simple but otherwise arbitrary statement as a norm (e.g., the rule is to put a ball in the blue area)."

## 1.1 Interindividual differences in rule-breaking

Although some people tend to break norms when the rewards of following them are limited, others always follow them regardless of the cost [16, 27]. However, in previous studies, unconditional rule-followers (i.e., participants who usually followed rules) were either excluded for further analyses [18, 23], or participants were directly instructed to follow or break rules (i.e., non-spontaneous rule-breaking behaviour; [24, 25, 28–32]). Thus, cognitive research on rule-breaking has rarely focused on this interindividual variability [18, 23]. Importantly, studying rule-breaking with in one individual is valuable and has been well studied [28–32], but distinct from a line of research that includes interindividual differences in rule-breaking. This gap is partly rooted in the challenge to design a paradigm in which participants show variable behaviour. On the one hand, the population divides into different groups with different tendencies towards imposed rules (i.e., rule-followers, rule-breakers; [18, 23]). Hence, it is difficult to study the natural inclinations of rule-following and rule-breaking within one individual. On the other hand, in a laboratory setting, it is difficult to induce spontaneous rule-breaking, which is not explicitly instructed and, thus, ecologically valid [22, 24, 25, 28, 29, 31, 33–35]. In tasks in which rule-breaking is not instructed, rule-breaking behaviour is substantially rare and it is, thus, hard to achieve the statistical power required to draw inferences on the comparison between breaking versus following rules. Research is needed to identify and describe the individuals' natural inclinations towards rule-breaking and to uncover their cognitive characteristics.

## 1.2 Spontaneous deliberative rule-breaking

One of the ways to motivate spontaneous deliberative rule-breaking in the laboratory is to introduce economic rewards [18, 36–41] so that participants are motivated to either increase their payoffs or to prevent losses [42]. In most studies, participants are encouraged to continuously break a rule [36–38, 40, 41]. However, in real-world situations, breaking a rule is not necessarily constant but merely sometimes optimal [43]. In many cases, rule-following might be beneficial and practical [31, 33, 44], while in other instances, rule-breaking may be the more advantageous option [45, 46]. Here, the self-interests of the individuals following or breaking

the rules are key. That is individuals' self-interests defined by an initial behaviour (i.e., the *a priori* behaviour expressed without an external constraint;[47–51]). In some individuals, this initial behaviour persists after a rule is present [23]. As rules limit or threat specific behavioural freedoms, individuals' psychological reactance might motivate to commit rule-breaking to pursue these self-interests [52]. They opt for committing rule violations only when the outcome is positive (e.g., leads to greater earnings or benefits), which is when it is aligned with their interests. For example, car drivers do not cross red traffic lights to reach their destination faster because someone or something commanded them to go for it [53]. This spontaneous and deliberative form of rule-breaking, i.e., the form of rule-breaking in which individuals carefully and meticulously decided whether to follow or break rules in particular situations based on consequences and own interests, is the one that is worth studying. However, research on this kind of spontaneous deliberative rule-breaking is rare and yet we need to comprehend the underlying cognitive mechanisms.

## 1.3 Cognitive conflict in rule-breaking

Previous literature on instructed rule-breaking elucidate the understanding of the cognitive characteristics of intentionally behaving contrary to what is commonly acknowledged as appropriate [25, 32, 54]. In these instructed rule-breaking tasks, when the individuals are asked to follow a rule, the rule retrieval automatically facilitates the agent's behaviour to obey the rule. Simultaneously, actions that are inconsistent with following the rule are suppressed. In contrast, when individuals are asked to break the rule, the individual must make the effort to reactivate covert actions or to look for alternative actions filling in for the behavioural option to follow the rule. Therefore, rule-breaking consists of resolving the cognitive conflict resulting from the simultaneous suppression of the rule-following action plan, alongside the intended action plan to break the rule [25, 55, 56]. The mere connotation of rule *violation* makes a response harder to carry out (e.g., take more time to complete) than an identical response that is labelled with a more neutral term, such as rule *inversion*, even if both actions require the same cognitive and motor operations [31, 33, 24].

Cognitive conflict can be measured and quantified by analysing reaction times and parameters of mouse trajectories such as Maximum Absolute Distance (MAD) and Area Under the Curve (AUC) in computerised tasks. For instance, if individuals have to choose between two options starting from a central point, then their mouse trajectory towards these options could determine the uncertainty towards one option or another [32]. Thereby, larger reaction times and larger trajectory parameters indicate larger cognitive conflict. Such measures are valuable because they (a) are sensitive to specific response options towards rules [57–59], (b) identify the temptation towards behavioural alternatives whilst probing for self-control [22], and (c) reflect internal representations such as anticipated action consequences [30, 33, 60].

Pfister *et al.* [18] and Wirth *et al.* [32] have suggested that, in instructed rule-breaking tasks, planning versus executing an action rely on separate cognitive processes. In typical conflict tasks, participants have to continuously react to task-relevant stimuli while ignoring task-irrelevant information. Responses are typically fast and correct but deteriorate once alternative responses are required [62–65]. However, with increasing frequency of alternative responses, participants' performance recovers [64–67]. Different from typical cognitive conflict tasks, the performance does not recover completely in instructed rule-breaking tasks [69]. Mainly, the time spent for planning rule-violations and not the execution of them remains unaffected by the frequency of the alternative responses (rule-breaking behaviour) and their recency (i.e., how often rule-breaking is immediately followed by further rule-breaking; [32]). This suggests that planning to violate norms is likely to involve persistent cognitive conflict. Evidence

implies, thus, that planning and execution of rule-breaking build upon different mental sources and processes [70, 71].

Recent research has shown that cognitive conflict relates not only to instructed rule-breaking, but also to spontaneous deliberative rule-breaking that requires to be an unsolicited but incentivised rule violation [18, 22, 39]. Like in instructed rule-breaking, Pfister *et al.* [22] observed that spontaneous deliberative rule-breaking relates to larger cognitive conflict than rule-following, and that this conflict was correlated with fewer decisions in favour of violating rules. Thus, individuals chose to violate norms even if it had a high cognitive cost. Nonetheless, few studies have examined cognitive conflict in spontaneous deliberative rule-breaking [18, 22], while several studies have examined this conflict in instructed rule-breaking [24, 25, 28–32]. Further studies are needed to evaluate this conflict in various tasks to confirm that this conflict is neither task specific nor due to the fact that this norm violation is instructed. In another spontaneous deliberative rule-breaking study, Arend [45] registered the frequency (i.e., how many times rule-breaking occurs in a given behavioural task to obtain a gain), latency (i.e., how early rule-breaking occurs in a given behavioural task to obtain a gain), and recency (i.e., how often rule-breaking is immediately followed by further rule-breaking resulting in a gain) of rule-breaking behaviour in a task. While latency was interpreted as individuals' alertness towards the recognition of opportunities, recency dictated the individuals' aggressiveness of their reaction towards positive feedback. However, Arend [45] did not explicitly consider cognitive conflict. He was not interested in the effect of the frequency, latency, and recency of rule-breaking behaviour on cognitive conflict, but on entrepreneurial status. Although frequency and recency of rule-breaking have been shown to impact the cognitive conflict during the execution but not the planning of rule violations in instructed rule-breaking tasks [32], how such results transfer to spontaneous deliberative rule-breaking remains unknown. Likewise, evidence on the influence of rule-breaking latency on cognitive conflict in spontaneous deliberative rule-breaking has been neglected.

In summary, research on instructed and spontaneous deliberative rule-breaking tasks has shown that rule-breaking involves cognitive conflict. This conflict is larger when breaking rules than when following them, and it can be measured via reaction times and mouse trajectory parameters. However, few studies have investigated this conflict in spontaneous deliberative rule-breaking in comparison to instructed rule-breaking. Moreover, the effect of frequency, latency, and recency on cognitive conflict in spontaneous deliberative rule-breaking has not been addressed.

## 1.4 DIMI Model

Previous rule-breaking studies have mainly focused on instructed rule-breaking behaviour [25, 29–31, 33, 35, 37]. Based on these studies, Wirth *et al.* [32] postulated the Decision-Implementation-Mandatory switch-Inhibition (DIMI) model, an adaptation of the two-step activation model [24]. The DIMI model assumes:

1. The following and breaking of norms are two distinct task sets, even when co-occurring in the same block. However, both task sets always co-occur [32].

2. By default, humans follow rules. Therefore, the task set for rule-following is always accessible and partially pre-implemented. This is evident because when participants obey norms, they take less time to complete this action.

3. When a rule-breaking task set is implemented, interference arises from the two task sets' competition (rule-breaking and rule-following) and triggers cognitive conflict (e.g., slower reaction times, complex and longer mouse movements). Here, the rule-following task set is

only temporarily suppressed or inhibited because the rule-breaking task set cannot exist independently.

4. The selection for the task set (i.e., rule-following versus rule-breaking) usually occurs mainly during its planning as it is evident due to slow reaction times [32]. On top of that, implementation of the task set is not necessarily complete by the end of its planning (i.e., initiation time, e.g., the time in which stimuli are displayed), but can continue even during the action execution (i.e., movement time, e.g., time in which participants perform movements to complete the task set), as it is evident due to slower reaction times in comparison to rule-following [72].

The DIMI model has been framed to explain instructed rule-breaking but has yet to conceptualise spontaneous deliberative rule-breaking. As cognitive conflict has already been observed in spontaneous deliberative rule-breaking behaviour [18, 24, 25], we expect the DIMI model to cover this behaviour as well. Empirical evidence for this hypothesis, however, still needs to be provided. Furthermore, the DIMI model has so far exclusively considered behaviours or task sets exclusively performed by the same individual. Although there are visible interindividual differences in rule-breaking [18, 23], previous research has not investigated the cognitive underpinnings of these differences for the challenges mentioned above (see *1.1 Interindividual differences*). Thus, the fit of interindividual differences in the DIMI model has not yet been explored. Since the model's assumptions do not exclude individuals who tend to follow the rules or commit spontaneous rule-breaking, we expect that the model applies to understand these interindividual differences. This hypothesis is yet subject to empirical support. If true, the model could contribute even more to the understanding of spontaneous deliberative rule-breaking because it would enable differentiation of the cognitive scheme in individuals that tend to commit rule-breaking from those who do not.

## 1.5 Personality in rule-breakers and rule-followers

Personality describes reasonably constant patterns of behaviour, thoughts, and emotions [73, 74] and accounts for a high amount of variance in various behavioural and cognitive processes [75–77]. However, the influence of personality on behaviours and cognitive processes has not been yet explored among rule-breakers and rule-followers in a controlled setting. For instance, personality could (a) facilitate low cognitive cost in rule-followers, (b) enhance the frequency of rule-breaking behaviour in rule-breakers, and (c) facilitate better coping with high cognitive costs due to spontaneous deliberative rule-breaking behaviour in rule-breakers.

Moreover, previous research has indicated personality to be a strong predictor of behaviours in individuals that conform or violate rules [75, 78]. For example, individuals that tend to conform to rules tend to be introverts [78]. In contrast, individuals who are inclined to break rules (e.g., criminal behaviour, pedestrian train crossing violation, pro-social rule-breaking, counterproductive behaviour at school, aggression) tend to be grandiose narcissistic [79], propense to risk [80–84], disagreeable [85–91], and impulsive (e.g., low behavioural inhibition, high goal-oriented motivation, and sensation seeking; [92–95]). Therefore, it seems worth to investigate the link of these personality traits (using psychometric measures) with the behavioural and the cognitive characteristics of rule-followers and rule-breakers, especially when evaluating these characteristics in a controlled setting.

## 1.6 Research goals

The present study aims to fill the outlined gaps by addressing the following five research goals:

1. As previous studies have rarely examined differential characteristics of rule-followers versus rule-breakers, in this study we implement and validate a computerised task that identifies interindividual differences in rule-breaking. While rule-followers tend to follow rules, rule-breakers tend to violate rules when the consequences of following them are negative, and tend to follow them when the consequences are positive. Rule-breakers pursue their self-interests as their initial behaviours persist even after external rules are imposed. This is important because we improve the characterisation of individual variations in responses towards rules and spontaneous deliberative rule-breaking as we appraise how individuals who commit this behaviour are distinct from others.

2. We aim to evaluate cognitive conflict in (a) rule-breakers versus rule-followers and (b) spontaneous deliberative rule-breaking versus rule-following in rule-breakers. This is important because (a) we improve the characterisation of individual variations in rule-breaking as we appraise how rule-breakers are distinct from others, and (b) we can specifically attribute conflict to spontaneous deliberative rule-breaking instead of instructed behaviour or the task in which this behaviour is tested.

3. Moreover, it is still unknown whether or not the factors such as frequency, latency and recency of rule-breaking impact cognitive conflict in spontaneous deliberative rule-breaking. In this line, we investigate to what extent this conflict is affected by them.

4. In the interest of providing a broad perspective on interindividual differences in rule-breaking, we investigate the relationships between personality, behaviour, and cognitive processes in rule-followers and rule-breakers.

5. In order to know whether the DIMI model extends (a) from instructed rule-breaking behaviour to spontaneous deliberative rule-breaking within and between individuals, and (b) from behaviours (rule-following versus rule-breaking) within an individual to behavioural tendencies (rule-followers versus rule-breakers) between individuals, we aim to discuss the extent to which this model fits our results. Framing our results in this model contributes to the conceptualisation of spontaneous deliberative rule-breaking.

## 2. Method

### 2.1 Sample and procedure

The study was conducted in either German or English in the Laboratory of Experimental Research in Economics at the Technical University of Munich. Once participants arrived at the laboratory, they signed a written informed consent and sat in individual cubicles to complete the computerised task and questionnaires. The entire experiment took about one hour to complete and at the end of the experiment participants were paid between 8 to 14 euros for compensation. All procedures were approved by the Ethics Commission of Technical University Munich (project number: 64/19 s).

A final sample of 133 participants (61 females, i.e., 45.9%; $M_{age}$ = 25, $SD_{age}$ = 7) were included in the analysis. In terms of outlier analysis, first, we excluded practice trials, trials that took longer than 5000 ms or shorter than 250 ms [96–98], and outlier values of the reaction times and mouse trajectory parameters during the "rule" part of the task, which resulted in the exclusion of .07% of trials. Second, following the main resistance rule by Hoaglin *et al.* [99], we performed an outlier analysis on the mean of the reaction's times and mouse trajectory parameters of the trials of all participants in the blocks where rules were shown, which led to the exclusion of one participant (for further details on outlier analysis and excluded participant see S2 File).

## 2.2 Rule-breaking task

To measure rule-breaking behaviour, we implemented a computerised task adapted from an established paradigm [24]. See *Fig 1* for task design, and an animation in.GIF format of the whole task can be found in the S2 File.

**2.2.1 Technical specificities.** Viewing distance was unconstrained at approximately 65 cm. Stimuli were presented on a 19-inch CRT monitor (1440 x 900 pixels, 75-Hz vertical refresh rate) and enhanced pointer precision in mouse settings was deactivated to obtain an accurate measurement of participants' mouse trajectory parameters (13-14Hz refresh rate). E-Prime 3.0 (Psychology Software Tools, Pittsburgh, PA) was used to implement the computerised experimental task.

**2.2.2 Instructions.** At the beginning of the task, participants were informed that they would receive 8 Euro for participation at the end of the experiment and that this amount would increase proportionally to the number of stocks they earned during the task. We read the initial instructions and asked the participants to reread these on the screen before they proceeded with the task. In addition, participants were instructed to execute smooth movements, and were encouraged to ask questions to ensure that they understood the task. Experimenters were present in the room during the whole procedure.

**2.2.3 General task procedure.** In the experiment, the participants decided how to allocate balls between two areas: a blue and an orange box. Each box was associated with a different number of stocks for each trial (see *Fig 1A*). Participants earned the number of stocks that they

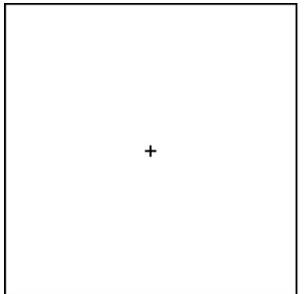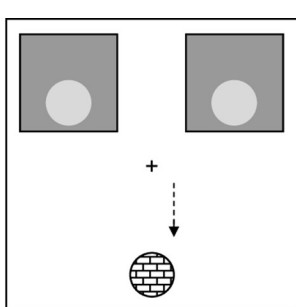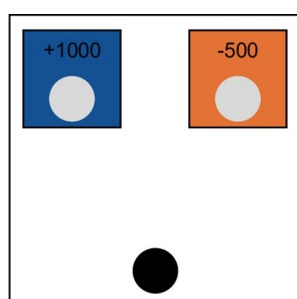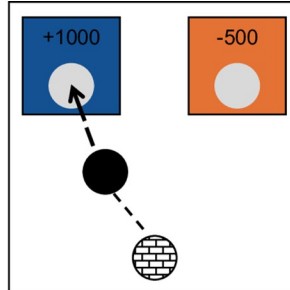

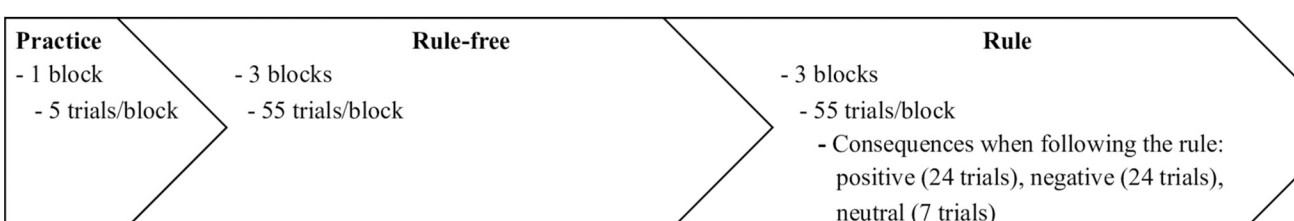

**Fig 1.** A. Trial structure. Following a fixation cross (500–700 ms), participants moved the mouse cursor to the home-area in the bottom-centre of the screen. Once they reached the home-area, the cursor transformed into a black ball. Simultaneously, the screen displayed the coloured boxes and the stock values for each box (the time prior to cursor pick-up while stock values were already displayed was measured as initiation time; i.e., action planning). Subsequently, participants dragged the ball from the home-area towards either of the boxes and, therewith, earned/lost the stocks associated with the chosen box (the time of the cursor movement was measured as movement time; i.e., action execution). For further information about the stimuli location on the screen see S2 File. B. Block structure. The task included a "rule-free" and a "rule" part, preceded by practice trials. After each block, participants received feedback on the total amount of stocks they accumulated. In "rule-free" blocks, participants were instructed to freely choose the number of stocks they wanted to keep for themselves. In the "rule" blocks, participants were instructed to select a specific colour, irrespective of the associated stocks. The type of consequences when following the rule in the "rule" part were fully randomised. The experiment comprised 335 trials in total.

selected. Participants' decisions throughout the task led to real financial consequences because the final sum of chosen stocks translated into additional compensation.

**2.2.4 Trial structure.** *Fig 1A* summarises the trial procedure. Trials commenced with a fixation cross of 500–700 ms duration (jittered randomly in steps of 20 ms). Afterwards, the following objects were presented on the screen: a brick black-white texture circle (diameter: 2 cm) in the lower part of the screen, a cross-shaped cursor (diameter = 2 cm) situated in the screen centre, two grey boxes located on the superior part of the screen that were separated horizontally by 16 cm and had inside a circular light grey hole (diameter = 2,2 cm) (see *Fig 1A*). In each trial, the cursor had to be dragged to the home-area (brick black-white texture circle) to pick up a black ball (diameter = 2 cm), which subsequently displayed at the respective cursor coordinates. At the same time, each box turned into either blue or orange, and a specific number of stocks appeared above the grey hole. The time spent in the home area while seeing the amounts of stocks displayed on the screen was registered as the initiation time (i.e., action planning). Then, the participants dragged the ball into one of the grey holes located in the boxes to complete the trial, which meant that they selected the number of stocks displayed above the chosen box. The time between when the ball was dragged out of the home-area and dropped into a hole was registered as the movement time (i.e., action execution). If participants took more than 1000 ms to complete this action, a message, *"Please try to leave the home-area as quickly as possible!"*, would appear on the screen so that participants became faster and remained focused throughout the task. The assignment of colours to each of the boxes was randomised across trials. The use of blue versus orange ensured that all participants recognised them as two different colours, even if they were colour-blind. For further information about the stimuli location on the screen see S2 File.

**2.2.5 Block structure.** The task consisted of two parts: an initial "rule-free" part and a subsequent "rule" part, both preceded by five practice trials (see *Fig 1B*). The "rule-free" part included three blocks in which the participants freely chose the box they preferred. The "rule" part involved three blocks and introduced a simple colour-based rule that was displayed on screen (e.g., "The rule is to put each ball in the blue/orange area") at the beginning of each block. The colour indicated in this rule was counterbalanced across participants. Rule-breaking did not have any additional consequence apart from receiving or losing the number of stocks associated with the chosen box. Each block included 55 trials. When following the rule during the "rule" part, 7 trials led to neutral (i.e., getting the same number of stocks), 24 to positive (i.e., getting the greatest number of stocks), and 24 to negative (i.e., getting the lowest number of stocks) consequences. The trial sequence within each block was fully randomised. An additional diagram about the block structure can be found in the S2 File.

**2.2.6 Decision consequences.** In each trial, dragging the ball to either the blue or the orange box led to the following consequences in terms of stock amounts: −5000, -3000, −1000, −500, 0, 500, 1000, 3000, 5000. Participants were confronted with combinations of these stocks across the two boxes within each block (see S2 File for further details of these combinations). The distribution of stocks was arranged in a way that unconditional rule-followers earned a maximum of 39000 stocks in the "rule" part, while rule-breakers always opting for the most beneficial option—despite the consequence when following the rule—could earn a maximum of 171000 stocks. This potential earning was implemented to increase the motivation to break rules.

## 2.3 Questionnaires

After participants completed the main task, personality traits were assessed via psychometric measurements. We evaluated narcissism using the Narcissistic Personality Inventory

(cronbach's alpha: .62; [100], 13 items, literature cronbach's alpha = .73), risk propensity (cronbach's alpha: .68; [101], 2 items, literature cronbach's alpha = .75), impulsiveness—behavioural inhibition and activation systems (cronbach's alpha = .7; [102], and the Big Five personality traits (cronbach's alpha = .45; [103], 10 items, literature cronbach's alpha = .75), for further details see S2 File.

### 2.4 Data analyses

**2.4.1 Classification of rule-breakers versus rule-followers.** All data analysis is based on classification of participants into two groups. Participants were classified as either rule-breakers or rule-followers based on the distribution of the percentage of rule-breaking behaviour that led to a gain. Participants who always follow the rule and who were within the first quantile (25%) of the distribution in individuals who broke at least once the rule were labelled as rule-followers, and the rest were labelled as rule-breakers. Thus, we opted for a conservative criterion regarding the inclusion of participants as rule-followers, by including within rule-followers individuals who never broke the rule and those who might have mistaken on breaking the rule (i.e., those in the first quantile). Importantly, including only those who always followed the rule led to the same results. In addition, we controlled that rule-breakers followed the rules in more than 95% of trials in which the consequences of following them were positive.

**2.4.2 Statistical analyses.** Mouse trajectory parameters were extracted from the raw movement trajectories during the movement time by using a custom-coded MATLAB (MATLAB 2019a, The MathWorks, Natick, 2019) based on Wirth *et al.* [32]. All data was processed in R version v3.1.2. and statistical analyses were performed in in IBM SPSS Statistics (Version 27). General linear models (2-tailed, sig. .05) were employed.

## 3. Results

### 3.1 Classification of rule-breakers versus rule-followers

Participants who never broke the rule or that broke the rule in less than or equal to 13.8% (first quantile cut-off; see also method section) were classified as rule-followers (N = 63, 30 females, i.e., 49.2%; $M_{age}$ = 25.4, $SD_{age}$ = 7.7), while the rest were labelled as rule-breakers ($N$ = 70, 31 females. i.e., 44.3%. $M_{age}$ = 25, $SD_{age}$ = 6.4). Importantly, most of the rule-followers always followed the rules regardless of the consequences (78.6%). Results were stable when using other cut-offs (0%, 5%, 10%, 15%, 20%, 55%)[1]. As assessed via manipulation checks, all participants reported that they recognised and remembered the rule in the "rule" part of the task, as well as had no previous experience with similar tasks. All rule-breakers explicitly reported that they sometimes broke the rules.

### 3.2 Rule-breaking task

**3.2.1 Decision-making in the "rule-free" and "rule" part.** Participants optimised their earnings in 97% of the trials in the "rule-free" part (rule-breakers = 97%; rule-followers = 95%), which shows that their intrinsic interest was to maximise their earnings. Participants were slower and exhibited longer and more complex trajectories in the "rule-free" part than in the "rule" part. This suggests that participants learnt to master the task after the "rule-free" part (see S2 File for details). Furthermore, rule-breakers' and rule-followers' reaction times and mouse trajectories exclusively differed significantly in the "rule" part of the task (see S2 File for details).

**3.2.2 Reaction times and mouse trajectories across the type of consequences when following the rule and across interindividual differences in rule-breaking.** Multiple

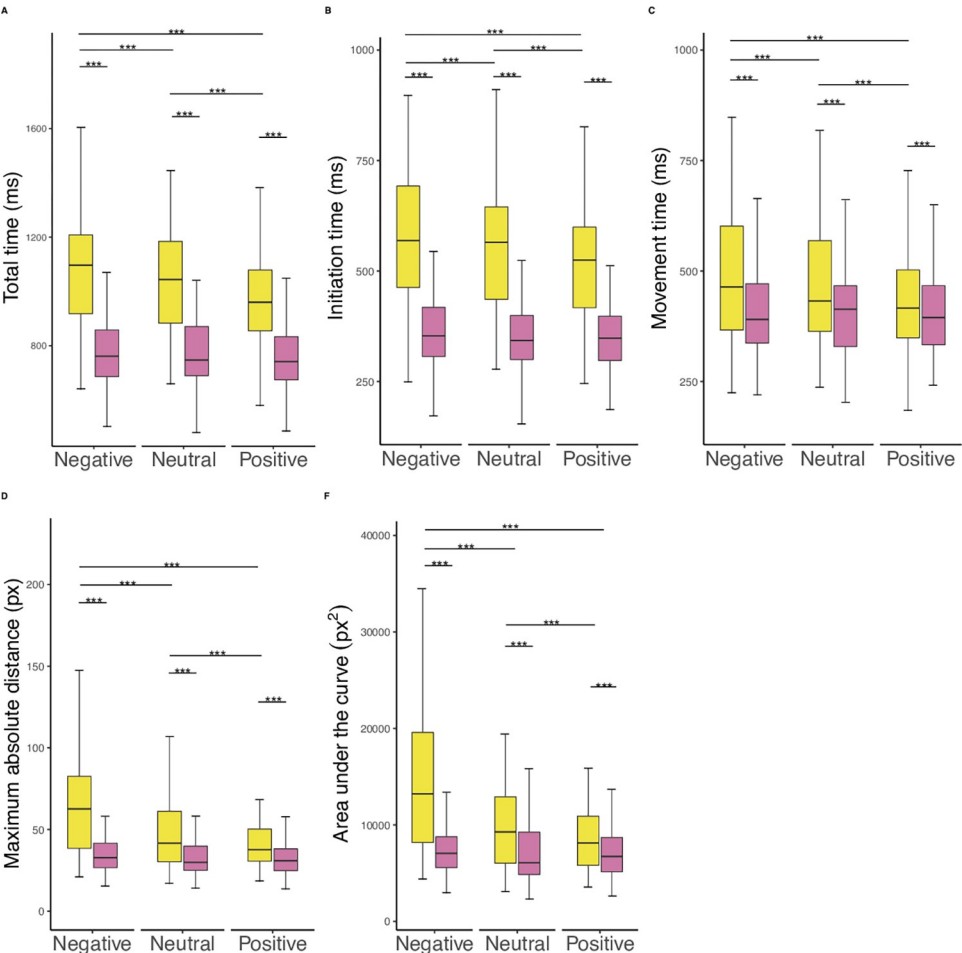

**Fig 2. Reaction times and mouse trajectories across interindividual differences in rule-breaking and type of consequences when following the rule.** Yellow = rule-breakers, Pink = rule-followers. Significance: *** = p < .001. The top and bottom whiskers are set to the highest/lowest value of the dataset that are included in the 1.5IQR range.

independent mixed 3 x 2 ANOVAs with the type of consequence when following the rule (i.e., positive, negative, neutral) as a within group factor and the behavioural tendency (i.e., rule-followers versus rule-breakers) as a between group factor were computed to examine whether there were significant differences in participants' behaviour (i.e., reaction times, mouse trajectory parameters) (see *Fig 2*). These tests revealed an interaction between the type of consequences and the behavioural tendency with respect to reaction times and mouse trajectory parameters ($F_{total(2,130)} = 33.23$, $\eta^2 = .34$, $p < .001$; $F_{initiation\,(2,\,130)} = 18.65$, $\eta^2 = .22$, $p < .001$; $F_{movement\,(2,\,130)} = 18.6$, $\eta^2 = .22$, $p < .001$; $F_{MAD\,(2,\,130)} = 34.95$, $\eta^2 = .35$, $p < .001$; $F_{AUC\,(2,\,130)} = 35.68$, $\eta^2 = .35$, $p < .001$). Rule-breakers were significantly slower, displayed longer, and more complex mouse trajectories across all type of consequences when following the rule as compared to rule-followers (see *Table 1*, *Fig 2*). Trials associated with negative consequences resulted in significantly slower reactions, as well as longer and more complex mouse trajectories than those associated with positive and neutral consequences (see *Table 1*, *Fig 2*). Further post-hoc analyses with Bonferroni adjustment revealed that rule-breakers' reactions were characterised by longer total, initiation, and movement time in trials associated with negative consequences as compared to when those associated with positive consequences (see *Table 1*, *Fig 2*). Mouse trajectories were longer and more complex in rule-breakers in trials associated with

**Table 1. Descriptive values and post-hoc results of reaction times and mouse trajectory parameters across behavioural tendency and type of consequences when following the rule (i.e., neutral, positive, negative).**

**Descriptive analyses**

| | Type of consequences | | | | | | | | | | | |
|---|---|---|---|---|---|---|---|---|---|---|---|---|
| | Negative | | | | Neutral | | | | Positive | | | |
| | Rule-breakers | | Rule-followers | | Rule-breakers | | Rule-followers | | Rule-breakers | | Rule-followers | |
| | Mean | SD | Mean | SD | Mean | SD | Mean | SD | Mean | SD | Mean | SD |
| Total time (ms) | 1098.1 | 219.3 | 787.7 | 168.6 | 1044 | 196.2 | 780 | 164 | 970 | 168.9 | 765.7 | 156 |
| Initiation time (ms) | 576.1 | 170.5 | 361.5 | 85.8 | 559.1 | 153.2 | 353.7 | 81.2 | 523.1 | 134.8 | 351.2 | 71.5 |
| Movement time (ms) | 522 | 216.2 | 426.1 | 134.5 | 484.8 | 198 | 426.3 | 133.9 | 446.9 | 168.6 | 414.4 | 128.7 |
| MAD (px) | 69.4 | 40 | 38.4 | 22.7 | 52.1 | 34.4 | 36.7 | 21.2 | 43.1 | 24.5 | 35.7 | 19.2 |
| AUC (px$^2$) | 15439.6 | 9223.4 | 8817.5 | 6776.5 | 11245.3 | 7910.4 | 8971.9 | 7054.1 | 9743.3 | 6613.9 | 8197.6 | 6091.6 |

**Post-hoc analyses**

| | | | Mean difference | Std. Error | p | 95% Confidence interval | |
|---|---|---|---|---|---|---|---|
| | | | | | | Lower bound | Upper bound |
| *Total time* (ms) | | | | | | | |
| | *Behavioural tendencies* | | | | | | |
| | Rule-breakers | Rule-Followers | 259.56* | 29.99 | 0 | 200.24 | 318.9 |
| | *Type of consequences* | | | | | | |
| | Negative | Positive | 75.04* | 7.75 | 0 | 56.25 | 93.84 |
| | | Neutral | 30.89* | 8.47 | 0 | 10.34 | 51.44 |
| | Positive | Neutral | -44.154* | 5.22 | 0 | -56.81 | -31.5 |
| | *Behavioural tendency*: *Type of consequence* | | | | | | |
| | Rule-breakers: Negative | Rule-followers: Positive | 332.44* | 31.15 | 0 | 240.45 | 424.42 |
| | | Rule-followers: Negative | 310.45* | 31.15 | 0 | 218.47 | 402.44 |
| | | Rule-followers: Neutral | 318.07* | 31.15 | 0 | 226.09 | 410.06 |
| | | Rule-breakers: Positive | 128.1* | 31.96 | 0 | 33.73 | 222.48 |
| | | Rule-breakers: Neutral | 54.16 | 31.96 | 1 | -40.22 | 148.53 |
| | Rule-breakers: Positive | Rule-followers: Positive | 204.33* | 31.15 | 0 | 112.35 | 296.32 |
| | | Rule-followers: Negative | 182.35* | 31.15 | 0 | 90.36 | 274.34 |
| | | Rule-followers: Neutral | 189.97* | 31.15 | 0 | 97.98 | 281.96 |
| | | Rule-breakers: Neutral | -73.95 | 31.96 | .32 | -168.32 | 20.43 |
| | Rule-breakers: Neutral | Rule-followers: Positive | 278.28* | 31.15 | 0 | 186.29 | 370.26 |
| | | Rule-followers: Negative | 256.3* | 31.15 | 0 | 164.31 | 348.28 |
| | | Rule-followers: Neutral | 263.92* | 31.15 | 0 | 171.93 | 355.9 |
| | Rule-followers: Negative | Rule-followers: Positive | 21.98 | 30.32 | 1 | -67.55 | 111.52 |
| | | Rule-followers: Neutral | 7.62 | 30.32 | 1 | -81.91 | 97.15 |
| | Rule-followers: Positive | Rule-followers: Neutral | -14.36 | 30.32 | 1 | -103.89 | 75.17 |
| *Initiation time (ms)* | | | | | | | |
| | *Behavioural tendencies* | | | | | | |
| | Rule-breakers | Rule-Followers | 197.27* | 20.34 | 0 | 157.04 | 237.5 |
| | *Type of consequences* | | | | | | |
| | Negative | Positive | 31.67* | 4.56 | 0 | 20.6 | 42.74 |
| | | Neutral | 12.42* | 4.77 | .03 | 0.87 | 23.98 |
| | Positive | Neutral | -19,250* | 3.25 | 0 | -27.12 | -11.38 |
| | *Behavioural tendency*: *Type of consequence* | | | | | | |
| | Rule-breakers: Negative | Rule-followers: Positive | 224.87* | 20.92 | 0 | 163.09 | 286.65 |
| | | Rule-followers: Negative | 214.51* | 20.92 | 0 | 152.73 | 276.29 |

(*Continued*)

**Table 1.** (Continued)

| | | Rule-followers: Neutral | 222.38* | 20.92 | 0 | 160.6 | 284.17 |
|---|---|---|---|---|---|---|---|
| | | Rule-breakers: Positive | 52.98 | 21.46 | .21 | -10.41 | 116.37 |
| | | Rule-breakers: Neutral | 16.97 | 21.46 | 1 | -46.42 | 80.36 |
| | Rule-breakers: Positive | Rule-followers: Positive | 171.89* | 20.92 | 0 | 110.1 | 233.67 |
| | | Rule-followers: Negative | 161.53* | 20.92 | 0 | 99.74 | 223.31 |
| | | Rule-followers: Neutral | 169.4* | 20.92 | 0 | 107.62 | 231.18 |
| | | Rule-breakers: Neutral | -36.01 | 21.46 | 1 | -99.4 | 27.37 |
| | Rule-breakers: Neutral | Rule-followers: Positive | 207.9* | 20.92 | 0 | 146.12 | 269.68 |
| | | Rule-followers: Negative | 197.54* | 20.92 | 0 | 135.76 | 259.32 |
| | | Rule-followers: Neutral | 205.41* | 20.92 | 0 | 143.63 | 267.2 |
| | Rule-followers: Negative | Rule-followers: Positive | 10.36 | 20.36 | 1 | -49.77 | 70.5 |
| | | Rule-followers: Neutral | 7.88 | 20.36 | 1 | -52.26 | 68.01 |
| | Rule-followers: Positive | Rule-followers: Neutral | -2.49 | 20.36 | 1 | -62.62 | 57.65 |
| *Movement time (ms)* | | | | | | | |
| | *Behavioural tendencies* | | | | | | |
| | Rule-breakers | Rule-Followers | 62.3* | 28.06 | .03 | 6.79 | 117.8 |
| | *Type of consequences* | | | | | | |
| | Negative | Positive | 43.37* | 5.37 | 0 | 30.36 | 56.38 |
| | | Neutral | 18.47* | 5.88 | .01 | 4.2 | 32.73 |
| | Positive | Neutral | -24,904* | 4.22 | 0 | -35.14 | -14.66 |
| | *Behavioural tendency: Type of consequence* | | | | | | |
| | Rule-breakers: Negative | Rule-followers: Positive | 107.57* | 28.69 | 0 | 22.83 | 192.31 |
| | | Rule-followers: Negative | 95.95* | 28.69 | .01 | 11.21 | 180.68 |
| | | Rule-followers: Neutral | 95.69* | 28.69 | .01 | 10.95 | 180.43 |
| | | Rule-breakers: Positive | 75.12 | 29.44 | .17 | -11.82 | 162.06 |
| | | Rule-breakers: Neutral | 37.19 | 29.44 | 1 | -49.75 | 124.13 |
| | Rule-breakers: Positive | Rule-followers: Positive | 32.45 | 28.69 | 1 | -52.29 | 117.18 |
| | | Rule-followers: Negative | 20.82 | 28.69 | 1 | -63.91 | 105.56 |
| | | Rule-followers: Neutral | 20.57 | 28.69 | 1 | -64.17 | 105.31 |
| | | Rule-breakers: Neutral | -37.93 | 29.44 | 1 | -124.87 | 49.01 |
| | Rule-breakers: Neutral | Rule-followers: Positive | 70.38 | 28.69 | .22 | -14.36 | 155.12 |
| | | Rule-followers: Negative | 58.76 | 28.69 | .62 | -25.98 | 143.5 |
| | | Rule-followers: Neutral | 58.5 | 28.69 | .63 | -26.24 | 143.24 |
| | Rule-followers: Negative | Rule-followers: Positive | 11.62 | 27.93 | 1 | -70.86 | 94.1 |
| | | Rule-followers: Neutral | -.25 | 27.93 | 1 | -82.73 | 82.22 |
| | Rule-followers: Positive | Rule-followers: Neutral | -32.45 | 28.69 | 1 | -117.18 | 52.29 |
| *Maximum absolute distance (px)* | | | | | | | |
| | Behavioural tendencies | | | | | | |
| | Rule-breakers | Rule-Followers | 17.91* | 4.37 | 0 | 9.26 | 26.56 |
| | Type of consequences | | | | | | |
| | Negative | Positive | 14.46* | 1.6 | 0 | 10.59 | 18.33 |
| | | Neutral | 9.49* | 2.16 | 0 | 4.25 | 14.72 |
| | Positive | Neutral | -4,971* | 1.5 | 0 | -8.61 | -1.33 |
| | Behavioural tendency: Type of consequence | | | | | | |
| | Rule-breakers: Negative | Rule-followers: Positive | 38.8* | 5.92 | 0 | 21.32 | 56.28 |
| | | Rule-followers: Negative | 36.12 | 5.92 | 0 | 18.65 | 53.61 |
| | | Rule-followers: Neutral | 38.28* | 5.92 | 0 | 20.8 | 55.76 |
| | | Rule-breakers: Positive | 31.34* | 6.07 | 0 | 13.4 | 49.27 |

(*Continued*)

**Table 1.** (*Continued*)

| | | | | | | | |
|---|---|---|---|---|---|---|---|
| | | Rule-breakers: Neutral | 20.39* | 6.07 | .01 | 2.46 | 38.33 |
| | Rule-breakers: Positive | Rule-followers: Positive | 7.46 | 5.92 | 1 | -10.02 | 24.94 |
| | | Rule-followers: Negative | 4.79 | 5.92 | 1 | -12.69 | 22.27 |
| | | Rule-followers: Neutral | 6.93 | 5.92 | 1 | -10.54 | 24.42 |
| | | Rule-breakers: Neutral | -10.94 | 6.07 | 1 | -28.88 | 6.99 |
| | Rule-breakers: Neutral | Rule-followers: Positive | 18,41* | 5.92 | .03 | 0.93 | 35.89 |
| | | Rule-followers: Negative | 15.73 | 5.92 | .12 | -1.75 | 33.21 |
| | | Rule-followers: Neutral | 17,88* | 5.92 | .04 | 0.4 | 35.36 |
| | Rule-followers: Negative | Rule-followers: Positive | 2.68 | 5.76 | 1 | -14.34 | 19.69 |
| | | Rule-followers: Neutral | 2.15 | 5.76 | 1 | -14.86 | 19.16 |
| | Rule-followers: Positive | Rule-followers: Neutral | -0.53 | 5.76 | 1 | -17.54 | 16.49 |
| *Area under the curve* (px$^2$) | | | | | | | |
| | *Behavioural tendencies* | | | | | | |
| | Rule-breakers | Rule-Followers | 3647.04* | 1207.32 | 0 | 1258.68 | 6035.41 |
| | *Type of consequences* | | | | | | |
| | Negative | Positive | 3158.11* | 351.04 | 0 | 2306.81 | 4009.42 |
| | | Neutral | 2269.96* | 533.89 | 0 | 975.21 | 3564.71 |
| | Positive | Neutral | -888.17 | 367.27 | .05 | -1778.84 | 2.52 |
| | *Behavioural tendency*: *Type of consequence* | | | | | | |
| | Rule-breakers: Negative | Rule-followers: Positive | 8356.54* | 1548.78 | 0 | 3782.6 | 9223.37 |
| | | Rule-followers: Negative | 7756.63* | 1548.78 | 0 | 3182.69 | 9223.37 |
| | | Rule-followers: Neutral | 8071.96* | 1548.78 | 0 | 3498.02 | 9223.37 |
| | | Rule-breakers: Positive | 6727.4* | 1589.02 | 0 | 2034.64 | 9223.37 |
| | | Rule-breakers: Neutral | 4769.46* | 1589.02 | .04 | 76.69 | 9223.37 |
| | Rule-breakers: Positive | Rule-followers: Positive | 8356.54* | 1548.78 | 0 | 3782.6 | 9223.37 |
| | | Rule-followers: Negative | 7756.63* | 1548.78 | 0 | 3182.69 | 9223.37 |
| | | Rule-followers: Neutral | 8071.96* | 1548.78 | 0 | 3498.02 | 9223.37 |
| | | Rule-breakers: Neutral | 4769.46* | 1589.02 | .04 | 76.69 | 9223.37 |
| | Rule-breakers: Neutral | Rule-followers: Positive | 3587.08 | 1548.78 | .32 | -986.86 | 8161.03 |
| | | Rule-followers: Negative | 2987.17 | 1548.78 | .82 | -1586.77 | 7561.12 |
| | | Rule-followers: Neutral | 3302.5 | 1548.78 | .50 | -1271.45 | 7876.44 |
| | Rule-followers: Negative | Rule-followers: Positive | 599.91 | 1507.47 | 1 | -3852.04 | 5051.86 |
| | | Rule-followers: Neutral | 315.33 | 1507.47 | 1 | -4136.62 | 4767.28 |
| | Rule-followers: Positive | Rule-followers: Neutral | -284.58 | 1507.47 | 1 | -4736.53 | 4167.37 |

Between and within subject factor main post-hoc results of the ANOVA 3x2 assuming independent groups. Post-hoc results remain when assuming dependence of the group, see S2 File. Additional post hoc results comparing all conditions (Behavioural tendency: Type of consequence) after performing a one-way ANOVA, see S2 File). Std. = standard.

negative consequences as compared to when those associated with positive or neutral consequences (see *Table 1*, *Fig 2*). In rule breakers, mouse trajectories were longer and more complex in trials associated with neutral consequences than with those associated with positive consequences (see *Table 1*, *Fig 2*). No significant differences were found in reaction times and mouse trajectory parameters among rule-followers across the type of consequences when following the rule (see *Table 1*, *Fig 2*).

Since we are interested in the interindividual differences of responses towards rules, we reported the results based on the dichotomous distinction between rule-followers and rule-breakers. Exploratory analyses that examined the continuous (versus dichotomous) effect of

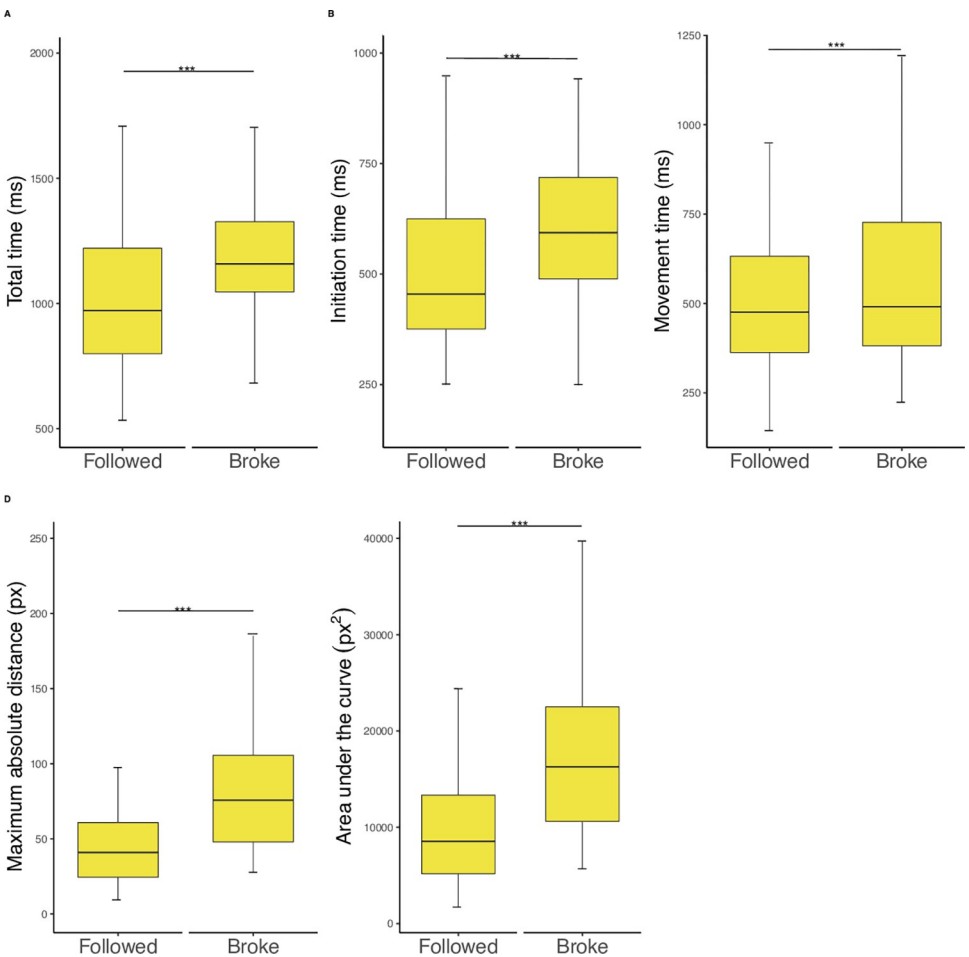

**Fig 3. Reaction times and mouse trajectories across responses to rules in rule-breakers.** Significance: $^{*}$ = p < .05, $^{**}$ = p < .01, $^{***}$ = p < .001. The top and bottom whiskers are set to the highest/lowest value of the dataset that are included in the 1.5IQR range.

rule-breaking frequency on reaction times and mouse trajectory parameters revealed the same results. All reported results remain stable after bootstrap analyses with 1000 permutations.

**3.2.3 Rule breakers.** *3.2.3.1 Reaction times and mouse trajectories when following and breaking the rules associated with negative consequences.* Multiple independent paired-sample t-tests were performed to examine the influence of the response towards the rule (i.e., rule-breaking behaviour versus rule-following behaviour) on cognitive conflict measurements (i.e., reaction times, mouse trajectory parameters) exclusively in those rule-breakers who sometimes broke and sometimes followed the rule in trials associated with negative consequences ($N$ = 59, 4 rule-breakers were excluded for this analysis as they always broke the rule in these trials). We found that rule-breakers were slower when they broke the rule ($M_{total}$ = 1183.3 ms, $M_{initiation}$ = 614.5 ms, $M_{movement}$ = 568.7 ms) than when they followed the rule ($M_{total}$ = 1028.8 ms, $M_{initiation}$ = 517.3 ms, $M_{movement}$ = 511.4 ms, $t_{total\ (56)}$ = -4.15, $\eta^2$ = -.55, $p$ < .001; $t_{initiation\ (56)}$ = -4.1, $\eta^2$ = -.58, $p$ < .001; $t_{movement\ (56)}$ = -4.4, $\eta^2$ = -.28, $p$ < .05). Notably, the effect is more pronounced for initiation than for movement time. Mouse trajectories were longer and more complex when breaking the rule ($M_{MAD}$ = 26 px; $M_{AUC}$ = 6549.3 px$^2$) than when following the rule ($M_{MAD}$ = 76.5 px, $M_{AUC}$ = 17016.8 px$^2$, $t_{MAD\ (58)}$ = -5.61, $p$ < .001; $t_{AUC\ (58)}$ = -5.37, $p$ <

**Table 2. The influence of frequency, recency, and latency of rule-breaking on reaction times and mouse trajectory parameters in the rule-part (rule-breakers, N = 63).**

| | Std. coeff. | Std. error | Beta coeff. | t | p | 95% Confidence interval | | $R^2$ | $R^2$ adjusted |
|---|---|---|---|---|---|---|---|---|---|
| | | | | | | Lower bound | Upper bound | | |
| *Percentage of rule-breaking* | | | | | | | | | |
| Total time (ms) | 3.26 | .72 | .5 | 4.53 | 0 | 1.82 | 4.69 | .25 | .24 |
| Initiation time (ms) | 2.05 | .61 | .39 | 3.34 | 0 | .82 | 3.27 | .16 | .14 |
| Movement time (ms) | 1.21 | .83 | .18 | 1.45 | .15 | -.46 | 2.88 | .03 | .02 |
| MAD (px) (ms) | .2 | .15 | .17 | 1.31 | .2 | -.11 | .51 | .03 | .01 |
| AUC (px²) (ms) | 41.80 | 39.86 | .13 | 1.05 | .3 | -37.91 | 121.5 | .02 | 0 |
| *Recency* | | | | | | | | | |
| Total time (ms) | 2.61 | .66 | .45 | 3.94 | 0 | 1.28 | 3.93 | .2 | .19 |
| Initiation time (ms) | 1.68 | .55 | .36 | 3.04 | 0 | .58 | 2.79 | .13 | .12 |
| Movement time (ms) | .12 | .14 | .11 | .88 | .38 | -.16 | .4 | .01 | 0 |
| MAD (px) (ms) | .12 | .14 | .11 | .88 | .38 | -.16 | .4 | .01 | 0 |
| AUC (px²) (ms) | 22.81 | 35.77 | .08 | .64 | .53 | -48.71 | 94.32 | .01 | 0 |
| *Latency* | | | | | | | | | |
| Total time (ms) | -4.82 | 1.6 | -.36 | -3.01 | 0 | -8.03 | -1.61 | .13 | .12 |
| Initiation time (ms) | -3.04 | 1.32 | -.28 | -2.3 | .03 | -5.68 | -.4 | .08 | .07 |
| Movement time (ms) | -1.78 | 1.74 | -.13 | -1.02 | .31 | -5.26 | 1.7 | .02 | 0 |
| MAD (px) | -.73 | .31 | -.29 | -2.35 | .02 | -1.35 | -.11 | .08 | .07 |
| AUC (px²) | -168.9 | 80.32 | -.26 | -2.10 | .04 | -329.51 | -8.29 | .07 | .05 |

Note: All analyses remained significant after bootstrapping with 1000 permutations (see S2 File). Std = standard, coeff. = coefficient.

.001; see *Fig 3*). Further analyses showed that reaction times are longer, as well as mouse trajectories are longer and more complex, when participants broke the rule in the current trial after following the rule in the previous trial, in comparison to trials were participants either followed the rule consecutively or followed the rule in the current trial after violating it in the previous trial (see S2 File for details).

All reported results remain stable after bootstrap analyses with 1000 permutations (see S2 File for details).

*3.2.3.2 Effect of the frequency, latency, and recency on spontaneous deliberative rule-breaking behaviour in the "rule" part*. In order to investigate the effect of frequency (percentage of number of trials rule-breaking occurs in a given behavioural task to obtain a gain), recency (percentage of rule-breaking occurring immediately followed by further rule-breaking resulting in a gain), and latency (number of trials preceding first rule-breaking to obtain a benefit) of rule-breaking behaviour on cognitive conflict, we performed multiple regression analyses on the independent impact of latency, recency, and frequency rule-breaking behaviour on reaction times and mouse trajectory parameters of rule-breakers during the "rule" part, see Table 2. Low latency, high recency, and high frequency of rule-breaking behaviour were positively related to longer reaction times, $p < .001$, and particularly recency and high frequency of rule-breaking related to longer initiation time, $p < .001$, see Table 2. In addition, low latency related to longer and more complex mouse trajectories, $p < .001$, see Table 2. When exploring the same analyses only in trials in which rules were associated with negative consequences, the same pattern was observed (see S2 File for details). In trials associated with negative consequences in which participants broke rules to obtain a benefit no significant differences were found (see S2 File for details). All reported results remain stable after bootstrap analyses with 1000 permutations (see S2 File for details).

### 3.3 Questionnaires

**3.3.1 Rule-breakers versus rule-followers.** In order to investigate whether or not certain personality is associated to "rule-followers" versus "rule-breakers", we performed Pearson correlations. Rule-breakers were more positively associated to grandiose narcissism, than rule-followers (see Table 3). No additional significant results were found.

**3.3.2 Rule-followers.** In order to provide a broad perspective on interindividual differences in rule-breaking we correlated personality traits and rule-breaking task related variables (e.g., reaction times and mouse trajectory parameters; see S2 File for correlation tables). In rule-followers, introversion was related to slow movement. Likewise, sensation seeking (subscale of the behavioural activation system) was negatively related to the total time, initiation time, and movement time per trial.

**3.3.3 Rule breakers.** In rule breakers, disagreeableness, goal orientation (sub-scale the behavioural activation system) and sensation seeking (a scale of the behavioural activation system) were related to larger payoffs, frequency of rule-breaking, and recency. Moreover, risk propensity tended to inversely affect movement time and mouse trajectory parameters. Furthermore, low behavioural inhibition was inversely related to initiation time. When we correlated the same variables exclusively in trials in which the consequences of following rules were negative, results remained in line with these findings (see Table 3). When correlating the outlined variables in trials in which the consequences of following rules were negative and participants broke these to increase their earnings, the effect of risk propensity on movement time and mouse trajectory parameters disappeared (see S2 File).

## 4. Discussion

In order to investigate the individual default tendencies towards norms, we implemented and validated a rule-breaking task sensitive to distinguish rule-breakers from rule-followers.

**Table 3. Main correlation findings of rule-followers versus rule-breakers and personality, and across individuals in these two groups during the "rule" part.**

| Variables | 1 | 2 | 3 | 4 | 5 | 6 | 7 | 8 | 9 |
|---|---|---|---|---|---|---|---|---|---|
| Rule-followers versus rule-breakers [b] | -.13* | -.06 | .06 | -.09 | -.05 | -.03 | .05 | .09 | .02 |
| *Rule-followers* | | | | | | | | | |
| Total pay-off | -.08 | -.1 | -.05 | -.04 | -.13 | .08 | .04 | -.03 | -.08 |
| Total time (ms) | -.11 | -.13 | .12 | -.12 | -.01 | -.04 | .31* | .2 | .09 |
| Initiation time (ms) | -.18 | .08 | .06 | .14 | .16 | -.04 | .22 | .2 | .1 |
| Movement time (ms) | -.14 | .11 | .1 | -.24* | -.11 | -.03 | .25* | .13 | .05 |
| AUC (px$^2$) | -.14 | .08 | .11 | -.04 | -.14 | -.02 | -.17 | -.17 | -.15 |
| MAD (px) | .13 | -.05 | .12 | -.07 | -.16 | .01 | -.13 | -.14 | -.15 |
| *Rule-breakers* | | | | | | | | | |
| Total pay-off | -.18 | -.36 | -.21* | -.2 | -.21* | .19 | .28* | .29* | .24 |
| Total time (ms) | 0 | -.36* | -.21 | -.15 | -.2 | -.17 | .27* | .26* | .18 |
| Initiation time (ms) | .02 | 0 | .07 | .16 | .13 | .02 | -.19 | -0.16 | -.08 |
| Movement time (ms) | -.02 | -.38* | -.19 | -.13 | -.16 | -.14 | .25* | .24* | .2 |
| AUC (px$^2$) | .03 | -.16 | .2 | .06 | -.24 | -.16 | -.1 | .14 | .01 |
| MAD (px) | .04 | -.05 | -.04 | -.02 | -.01 | -.26* | .04 | .07 | -.01 |

**Note:** 1 Grandiose narcissism, 2 Agreeableness, 3 Conscientiousness, 4 Extraversion, 5 Risk propensity, 6 BAS drive, 7 BAS fun seeking, 8 BAS reward, 9 BIS.

AUC = area under the curve, MAD = Maximum absolute distance

[b] 1 = rule-followers. 2 = rule-breakers. Correlation significance is at the .05 level (2 –tailed) represented with asterisk (*). Descriptive of the variables and further correlation analyses can be found in the S19-S23 Tables in S2 File.

Because rule-breakers are characterised by deliberatively violating norms that match their interests, they exclusively broke rules when these actions led to higher payoffs. Rule-breakers obtained higher earnings and exhibited higher cognitive conflict (i.e., slower responses, longer, and complex mouse trajectories), compared to rule-followers. Rule-breakers also exhibited higher cognitive conflict when the consequences of following the rules were negative than when they were either neutral or positive. In those trials associated with negative consequences (i.e., following the rules leads to limited rewards or losses), rule-breakers experienced more cognitive conflict when they broke the rules compared to when they followed them. Notably, cognitive conflict during action planning of rule-breaking behaviour was more pronounced than during action execution. In the "rule" part and in trials associated with negative consequences, the cognitive conflict experienced during action planning by rule-breakers was enhanced by low latency, high frequency, and high recency of rule-breaking. However, this effect disappeared when analyses were focussing trials in which rule-breakers violated rules. Moreover, in rule-followers, introversion and sensation seeking were associated with fast responses. In contrast, in rule-breakers, disagreeableness, sensation seeking, and goal-oriented motivation were associated with higher payoffs, frequency, and recency of rule-breaking. In parallel, in rule-breakers, risk propensity and behavioural inhibition were associated with fast planning and execution of the actions in the "rule" part of the task.

## 4.1 Broading the Decision-Implementation-Mandatory Switch-Inhibition model

Studies in which the DIMI model has been applied [32] focused on instructed rule-breaking and intraindividual differences (i.e., rule-breaking and rule-following within a single individuum). However, our investigation evaluated spontaneous deliberative rule-breaking, intraindividual differences (i.e., rule-breaking versus rule-following in rule-breakers), and interindividual differences (i.e., rule-breakers versus rule-followers). Although our study approach towards studying rule-breaking differs from previous work, our results fit and, thus, extend the DIMI model. Based on our results, the DIMI model explains (a) instructed as well as *spontaneous deliberative rule-breaking* and (b) intraindividual as well as *interindividual differences*. In the following paragraphs we explain how our results strongly support the assumptions of the DIMI model.

*First and second assumptions: Rule-following and rule-breaking responses rely on two distinct task sets and the rule-following task set is partially pre-implemented.*

These assumptions are supported by our findings because, in our task, rule-breakers switched between rule-breaking and rule-following task sets within the same block. Rule-followers were faster than rule-breakers, and the latter were faster when following rules than when breaking them. Therefore, the rule-following task set is different from the rule-breaking task set, but the former was partially pre-implemented–independent of whether both behaviours are displayed by one single or distinct agents.

*Third assumption: cognitive conflict occurs due to interference from the simultaneous activation of the two task sets.*

Our results support this assumption as cognitive conflict was more pronounced (a) in rule-breakers when comparing rule-breaking to rule-following in trials associated to negative consequences (i.e., intraindividual differences) and (b) in rule-breakers versus rule-followers (i.e., interindividual differences). Different from instructed rule-breaking tasks, rule-breakers in our task switched between task sets spontaneously (i.e., without explicitly giving them this instruction). Hence, they made this switch deliberatively to pursue an internal motive (i.e., to increase their earnings). In contrast, rule-followers constantly presented less cognitive conflict

because they continuously practiced rule-following and, thus, did not need to handle both task sets concurrently. Our data demonstrate that rule-following is associated with less cognitive conflict than spontaneous deliberative rule-breaking–independent of whether both behaviours are displayed by one single or distinct agents.

*Fourth assumption: (a) The selection of the task set occurs during action planning, (b) the implementation of the task set starts with action planning and lasts till action execution.*

The idea that the selection of a task set occurs early in the decision process is braced by electroencephalography research in instructed rule-breaking, in which the cognitive conflict present in the selection of the task set (rule-following versus rule-breaking) is reflected by a delayed and attenuated P300 component [31]. According to our data, the complex decision of choosing between rule-breaking versus rule-following task sets occurring during action planning took more cognitive effort and consequently more time than the subsequent action execution in rule-breakers. Therefore, the selection of the task set occurred early and recruited more cognitive sources than the execution of the action. The fourth assumption of the DIMI model is considered a signature of instructed rule-breaking. We can now extend this assumption to spontaneous deliberative rule-breaking.

Alternatively, the fourth assumption of the DIMI model applies to cognitive motor-control tasks. Kaiser *et al.* [104] found that cognitive conflict was more pronounced in action planning than in action execution during a motor-control task; this finding was mirrored by lower mid-frontal theta brain waves. The authors specify that this effect is due to the selection of motor responses, and that this effect is specific to cognitive motor control and does, for instance, not apply to attentional control. In typical cognitive-motor control tasks, participants repetitively perform a motor response (e.g., clicking a key repetitively) and inhibit this response when a signal is displayed on the screen. In our task, when participants committed rule-breaking, they inhibited the motor response that corresponds to rule-following. Thus, this inhibition process that appears in rule-breaking tasks seems similar to typical motor-control tasks.

How early, often, and recent rule-breaking occurred increased the cognitive conflict during the action planning, while no impact of these variables was observed on action execution in our data. This increment was observed when there was a switch between rule-following and rule-breaking task sets, and not when examining exclusively trials in which rule-breakers opted for the rule-breaking task set. Intuitively, one would expect that lower latency, higher frequency, and higher recency ameliorate the cognitive conflict related to this switching [64, 68]. However, these factors have shown, in instructed rule-breaking, to decrease cognitive conflict during action execution in rule-breaking, but not in action planning [32]. Therefore, cognitive conflict during action planning, seems to be more resistant to low latency, high frequency, and high recency of rule-breaking–most probably due to the selection of the task set.

The fourth assumption also advocates that the implementation of the task sets prolongates to its execution which indicates that cognitive conflict is present during action execution, but in a lower degree than in action planning. This assumption matches our results because during action execution cognitive conflict was more intense in (a) rule-breakers than in rule-followers and (b) in rule-breaking than in rule-following within rule-breakers. In contrast to previous research in instructed rule-breaking, the cognitive conflict related to action execution was not reversed by the recency and frequency of rule-breaking [32]. The absence of this effect in our study might be explained by limited frequency and recency of rule-breaking behaviour. The necessary threshold to ameliorate the costs of cognitive conflict in the execution of the task sets was probably not reached in our computerised task [18, 64, 68]. Further studies should evaluate spontaneous deliberative rule-breaking with paradigms optimized for the observation of frequency and recency. For instance, future studies might increase the number of blocks in

the "rule" part or increase the number of chances per block in which breaking the rule leads to gain.

## 4.2 Personality traits underlying interindividual differences in rule-breaking

The current study provides a broad perspective on the interindividual differences in rule-breaking by investigating how personality links with behaviours and cognitive processes in rule-breakers and rule-followers. Analogous to previous literature, we found that grandiose is more pronounced in rule-breakers than rule-followers [105]. Grandiose narcissism is characterized by self-importance, feelings of superiority, as well as exhibitionism [100, 106]. Indeed, narcissistic leaders are susceptible to violate norms as they are more likely to be innovative, but also to engage in unethical rule-breaking [79]. Moreover, grandiose narcissism has also been related to pro-social rule-breaking because individuals who are narcissists have the psychological need for grandiose fantasy, sacrificing, self-enhancement, and devaluing rules [105]. Narcissistic individuals believe that there are no limits to achieve their goals and that they are in control of their destiny, which makes them more prompt to violate norms to benefit themselves [107–109]. Grandiose narcissism has been shown to predict self-report measurements of proactive and reactive aggression, as well as actual aggressive behaviours [110]. Narcissists likely feel entitled to break rules when rules don't benefit them [107]. Therefore, our results match previous theory, suggesting that narcissism is more likely to be associated with rule-breakers rather than with rule-followers.

Regarding rule-breakers, disagreeableness was associated with frequent and repetitive infractions, consistent with previous research [86, 111–113]. A reason for this finding is that disagreeable individuals tend to break rules because they are less likely to regulate themselves, which deters them from recruiting attentional resources to obey the rules and ignore their natural impulses [90]. For instance, trait agreeableness has been positively associated to the adherence to rules for prevention of COVID-19 during the first 1.5 years of the pandemic. Individuals might adhere to the COVID-19 preventive rules because they tend to care for others and avoid conflicts, even when they believe that the danger of this disease is exaggerated (or even faked). This suggests that agreeableness might be a critical personality trait in mitigating the effect of a negative attitude towards the preventive measures and, thus, enhancing actual preventive behaviour.

Moreover, sensation seeking and goal-orientation in rule-breakers were related to recurrent and frequent norm violation. These characteristics might enhance the ability of rule-breakers to cope with the "cognitive pain" associated with the violation of norms, allowing them to break the rules more often [93, 95, 114, 115]. In contrast, sensation seeking in rule-followers was associated with fast responses, which suggests that they experienced low cognitive conflict. This is not surprising because rule-followers prefer to obey norms and generally conform rapidly [116]. Moreover, introvert rule-followers were faster in their responses than extrovert rule-followers. This was expected as introverted individuals tend to agree faster with others' opinions than extroverted individuals [78].

Additionally, our results showed that low behavioural inhibition in rule-breakers was associated with time-consuming action planning (i.e., more cognitive conflict during action planning). Low behavioural inhibition associates with sensitivity to punishment [117]. In our task, following the rules could constantly lead to loss of earnings; as such rule-breakers characterised by behavioural inhibition might effortfully choose to violate norms based on their aversion to losing [70, 118]. Upon action planning, rule-breakers decreased their cognitive conflict, which favoured smooth action execution. In our study we found that rule-breakers

who are prone to risk execute their actions rapidly. This is in line with previous research, as individuals who are more propense to risk seem to have the advantage of decreasing their cognitive conflict as exhibited by fast responses during the execution of their actions, which in turn makes them more susceptible to commit rule-breaking [80–84]. Interestingly, entrepreneurial status has been associated with individuals who are propense to risk and commit moderate rule-breaking (i.e., delinquency, offences in family and school) and not extreme rule-breaking (i.e., breaking an official contact, drug use, and crime). As entrepreneurs tend to face moral dilemmas in their work (e.g., choosing between overpromise or telling the truth of the current financial condition of their enterprise to convince investors, employees, and customers to support their endeavour), it is possible that having high risk propensity declines their cognitive conflict for committing rule-breaking, so they can afford to go around these dilemmas.

Overall, our findings advance the understanding of interindividual differences in rule-breaking. Our study exhibits how personality relates to the cognitive and behavioural characteristics of rule-breakers and rule-followers. Future research needs to explore these relationships in various rule-breaking tasks to further understand whether these findings transfer to other tasks, so to comprehend the mindset favouring conformism to rules or attenuating the cognitive conflict associated with spontaneous deliberative rule-breaking.

## 4.3 Limitations

While the evaluation of spontaneous deliberative rule-breaking is more ecologically valid than instructed rule-breaking, it comes with its own pitfalls. First, the distinction between participants intentionally violating rules versus error could be called into question. Even if participants reported intentional rule-breaking, this testimony is nothing but retrospective. We aimed at minimizing this issue by implementing a task with low difficulty for participants to avoid errors [23, 119]. What is more, rule-breakers broke the rule mostly when following the rule involved negative consequences and explicitly report their intention to break rules to obtain benefits. Therefore, we are confident that participants committed indeed intentional rule violations rather than simple slips or errors. Second, not all the participants committed rule-breaking with the same frequency. Therefore, we cannot generalise that rule-breakers who violate rules at a certain frequency ameliorated or intensified their cognitive conflict. Instead, we constrained ourselves to treat the frequency of rule-breaking in rule-breakers as a continuous variable to examine its effect on cognitive conflict. In our experiment, we did not find that rule-breaking frequency lead to ameliorate cognitive conflict. However, this could have been because of lack of power corresponding to specific frequencies of spontaneous deliberative rule-breaking. Higher frequencies of rule-breaking can reverse cognitive conflict [32, 35], which our study cannot speak to. Further studies should increase the power of specific rule-breaking frequencies to understand how this shape cognitive conflict, e.g., by extending the number of trials that tempt participants to commit spontaneous deliberative rule-breaking.

To pursue our fourth research goal, we explored the relationships between personality, behaviour, and cognitive processes in rule-followers and rule-breakers. Surely, pursuing of this goal implies alpha-error accumulation, as it necessitates multiple correlation analyses [120]. Another limitation of this study is that we studied basic general norms instead of more complex norms, such as social or legal laws. Our findings, thus, cannot be generalised to the latter. Moreover, individuals can choose to break some norms while they decide to follow others. Future studies should address intraindividual differences of rule-breaking in relation to rule type (e.g., simple versus complex rule; general versus specific rules; social versus non-social rule). Our research reveals interindividual differences when it comes to breaking general norms which can be motivation to investigate the very same differences in the context of other

types of consequence-dependent rules. Furthermore, we have focused mainly on evaluating the behaviour and personality traits associated with rule-breaking rather than ethical judgements. Nevertheless, previous studies on "perverse norms" (i.e., uncertain and unfulfilled norms imposed by members of the own group or an external agent) have shown that ethical judgements vary depending on the type of norm. Compared to situations in which the norm is crystal clear, individuals judging situations involving "perverse norms" tend to attribute a lack of trust and prestige to the agents imposing the rule and are less judgemental of those who violate the rules [121]. Future studies should examine the importance of own ethical judgements in the context of spontaneous deliberative rule-breaking behaviour and trust and prestige judgements of the agent administrating the rule. We believe this to be a piece of valuable information to understand (a) the similarities of our rules (i.e., simple, straightforward and arbitrary rules that can lead to loss or gain) with perverse norms and (b) the role of ethical judgements in motivating spontaneous deliberative rule-breaking. An additional limitation is that our sample consisted of mostly young participants, therefore our findings cannot be generalised across different age groups. Future studies should investigate rule-breaking across the lifespan.

## 4.4 Conclusion–interindividual differences in rule-breaking research

Although previous research has already disclosed that the same stimuli raise differential individual behavioural responses or tendencies towards rules [18, 23], our study is pioneering a detailed analysis of these tendencies. Previous studies excluded participants who tend to follow rules from data analyses which we included. Rule-breakers expressed higher cognitive conflict than rule-followers, not only when breaking but also when following the rule. While rule-breakers prioritise increasing their payoffs over high cognitive conflict, rule-followers prioritise low cognitive conflict over increasing their payoffs. Thus, we uncovered a trade-off between deliberatively deciding whether to follow or violate norms to obtain more earnings versus experiencing higher cognitive conflict. These results suggest that–as expected–in the long run, conformism is more cognitively efficient than adaptively switching between rule-breaking and rule-following. This might explain why people stop questioning rules–often shortly after they start following them. Perhaps, this study can give a first hint on why members of political parties, religious, or other authoritarian groups respect the rules of these institutions and stop questioning them in the long run, even when these rules negatively affect their surroundings [9, 122–125]. Recently the understanding of individual differences in rule-breaking became even more relevant as–due to the COVID-19 pandemic–the individual's adherence to infection mitigation measures affects others' lives daily [50, 85, 87, 111]. This study leaves the door open to genuinely investigate personal trends towards external regulations.

How rule-breaking as an individual tendency towards norms favours behaving "right" or "wrong" concerns morality and remains an open question. Rule-breaking can lead to adverse outcomes (e.g., legal problems, scientific misconduct, aggression; [126–132]), but it also has advantages [133]. Examples of positive consequences of rule-breaking include being seen as a person with moral courage [134, 135], good heart (e.g., nurses helping patients even when going against clinic statements; [136]), being creative [137], or becoming an entrepreneur (e.g., increasing your earnings by selling new products that overcome the established rules in the market; [138]). Further studies about spontaneous deliberative rule-breaking rather than instructed rule-breaking could offer insights into motivating this behaviour. Considering how frequency, recency, and latency of rule-breaking can affect cognitive conflict could target ways in which spontaneous deliberative rule-breaking does not imply an immense cognitive cost.

Such evidence might in the future enable fostering constructive or societally productive forms of rule-breaking.

In summary, the present study shows that there are interindividual differences in rule-breaking. While some individuals tend to follow the norms, others tend to violate them to obtain benefits at the cost of more cognitive conflict. Rule-breakers suppress rule-following tendencies that are automatically activated upon encountering rule-related stimuli, especially when they plan to violate norms. Therefore, cognitive conflict is a robust and reliable downstream consequence of spontaneously and deliberatively violating rules in rule-breakers, mainly during action planning. These findings support the DIMI model and broad the application of this model to the interindividual differences in rule-breaking, and particularly in spontaneous deliberative rule-breaking. Furthermore, certain personality traits relate and contribute to the understanding of behavioural and cognitive processes experienced by rule-followers and rule-breakers. Future studies should further investigate how personality and manipulations of latency, recency, and frequency of rule-breaking could ameliorate cognitive conflict in spontaneous deliberative rule-breaking and thus favour this behaviour. This research sheds light on the cognitive and personality characteristics of the interindividual differences in responses towards rules, and especially of spontaneous deliberative rule-breaking.

## Supporting information

**S1 File. This are the highlights of the article.**
(DOCX)

**S2 File. This is the supplementary material of the article.**
(DOCX)

**S1 Video. This is the GIF of the task.**
(MP4)

## Author Contributions

**Conceptualization:** Leidy Cubillos-Pinilla, Franziska Emmerling.

**Data curation:** Leidy Cubillos-Pinilla.

**Formal analysis:** Leidy Cubillos-Pinilla.

**Funding acquisition:** Franziska Emmerling.

**Investigation:** Franziska Emmerling.

**Methodology:** Leidy Cubillos-Pinilla.

**Project administration:** Leidy Cubillos-Pinilla.

**Resources:** Franziska Emmerling.

**Software:** Leidy Cubillos-Pinilla.

**Supervision:** Franziska Emmerling.

**Validation:** Leidy Cubillos-Pinilla.

**Visualization:** Leidy Cubillos-Pinilla.

**Writing – original draft:** Leidy Cubillos-Pinilla.

**Writing – review & editing:** Leidy Cubillos-Pinilla, Franziska Emmerling.

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
