## [Decision Letter · Decision Letter 0]

7 Jun 2022

PONE-D-22-11962Taking the chance! – Interindividual differences in rule-breakingPLOS ONE

Dear Dr.  Cubillos-Pinilla,

Thank you for submitting your manuscript to PLOS ONE. After careful consideration, we feel that it has merit but does not fully meet PLOS ONE’s publication criteria as it currently stands. Therefore, we invite you to submit a revised version of the manuscript that addresses the points raised during the review process.

As you can see below, one of the reviewers is more enthusiastic about the results than the other. Please, go through the comments of both reviewers and try to answer their points accurately. If you refrain from following a suggestion, please explain why you decided that way.

We look forward to receiving your revised manuscript.

Kind regards,

Jaume Garcia-Segarra

Academic Editor

PLOS ONE

Journal Requirements:

3. Please ensure that you include a title page within your main document. We do appreciate that you have a title page document uploaded as a separate file, however, as per our author guidelines (http://journals.plos.org/plosone/s/submission-guidelines#loc-title-page) we do require this to be part of the manuscript file itself and not uploaded separately.

"This work was supported by the funding of European Commission under the MARIE SKŁODOWSKA-CURIE ACTIONS Individual Fellowships (IF) grant, Call: H2020-MSCA-IF-2017."

"FE (Dr. Franziska Emmerling)

MARIE SKŁODOWSKA-CURIE ACTIONS Individual Fellowships (IF) grant, Call: H2020-MSCA-IF-2017

Link: https://ec.europa.eu/info/funding-tenders/opportunities/portal/screen/opportunities/topic-details/msca-if-2017

Reviewers' comments:

Reviewer's Responses to Questions

**Comments to the Author**

1. Is the manuscript technically sound, and do the data support the conclusions?

Reviewer #1: Yes

Reviewer #2: Yes

2. Has the statistical analysis been performed appropriately and rigorously? 

Reviewer #1: Yes

Reviewer #2: Yes

3. Have the authors made all data underlying the findings in their manuscript fully available?

Reviewer #1: Yes

Reviewer #2: No

4. Is the manuscript presented in an intelligible fashion and written in standard English?

Reviewer #1: Yes

Reviewer #2: Yes

5. Review Comments to the Author

Reviewer #1: The manuscript entitled “Taking the chance! – Interindividual differences in rule-breaking”. This manuscript investigates the cognitive characteristics of individuals who commit spontaneous deliberative rule-breaking (rule-breakers) versus rule-followers.

This is a topic of great interest and little studied, for which the authors are congratulated. In addition, the authors are congratulated because they have made a very adequate theoretical exposition of the topic (the authors are congratulated specifically for the explanation of cognitive conflict and the ways to measure it), methodologically it is a well-done job, the data analysis is correct, and the conclusions are of interest.

However, all work can be improved, so this reviewer wishes to point out a series of issues that can improve the manuscript:

1 Introduction:

I think that the meaning of the norm concept should be broadened. I think it would be interesting to distinguish between norms in general, social norms. legal norms and moral norms.

There is a concept, that of the perverse norm, which I think should at least be mentioned. See, for example, Oceja, L.V., and Fernandez-Dols, J.M.

I think that when dealing with the issue of breaking norms, it should be linked to that of psychological reactance. This perspective can be linked with antisocial behavior.

I think the cultural dimension should be taken into account. There are cultures in which social norms are more respected, and cultures in which they are less respected. I do not know bibliography about it, but if there is it would be interesting to mention this fact. Perhaps this distinguishes individualistic from collectivist cultures (studies by Hofstede, Schwartz, etc.)

1.1. Interindividual differences in rule-breaking

One could refer to the law as a coercive element of socialization.

The difficulty of studying rule-breaking within one individual is established, and it is true. But it should be insisted that it is another line of research, and that respect for some rules and breaking of others occur simultaneously in all subjects. This aspect should also be included in the limitations of the study.

The above shows, by exposing an intra-individual vision (cognitive aspects) that it would be logical to approach an intra-individual perspective.

1.4 DIMI Model (Decision-Implementation-Mandatory switch-Inhibition)

This model shows that the same subject can choose to skip the rules at a given time or follow them, that is, it serves to explain an intra-individual decision-making, and also allows determining the existence of two groups of subjects. It should be explained why this intra-individual perspective is not adopted.

1.5 Personality in rule-breakers and rule-followers

I think that works should be found that refer to impulsiveness.

2.1.

Better put participants

The median age is young teens (Mage = 25. SDage = 7); must be recognized in the limitations.

Questionnaires: Questionnaires: why those? Expand information on each questionnaire, put reliability and validity data from other studies, and at least reliability in this sample. Why use the Narcissism NPI quiz and not use a Dark Personality quiz in general? The Big Five is a very long questionnaire and is rarely used at work; Why this quiz?

Data analysis: specify the version of SPSS used.

RESULTS

3.1 Classification of rule-breakers versus rule-followers. How is it possible that the first quartile groups more or less half of the subjects, and the other three quartiles another half? I consider it statistically impossible. Furthermore, wouldn't it be better to create two extreme groups, the subjects that are grouped in the first quartile (rule breakers) versus those in the fourth quartile (rule followers)?

The title of the Tables must be placed at the top of the Tables, not at the bottom. In addition, “Note” must be added and the meaning of abbreviations must be specified, such as SD, Std. Error,*, **, etc.

The results referring to questionnaires should be accompanied by the respective Tables. It gives the impression that this part is not considered relevant.

DISCUSSION

The results are claimed to confirm the DIMI model, but I think a more detailed explanation is needed. I believe that the explanation that is elaborated does not coincide with the hypotheses that were specified before carrying out the investigation.

I think that the discussion should refer to all the variables of the study. What has happened, for example, with the Big Five variables? And with the NPI test?

Reviewer #2: Overall, the paper is well written. The research gap is well explained in the introduction, and the hypothesis are clear and interesting. The experiment and analyses seem to be well conducted. Some clarifications are required before the paper can be accepted, but these can probably be well addressed in a revision.

In 1.5, the authors list personality traits that are typically found to be related to rule-breaking. I was a bit surprised that some obvious characteristics such as narcissism, impulsiveness, honesty or Machiavellianism were not mentioned. Narcissism was then assessed in the study. A reference should also be added for narcissism in 1.5, or are there no previous studies about this? I would also appreciate if the authors briefly define the assessed concepts (narcissism, inhibition etc.) and explain why these concepts are related to rule-breaking.

Rule-breaking is of special interest during COVID-19 (e.g., not wearing masks, not going into quarantaine after being tested positive), and studying individual differences in adherence to infection mitigation measures is thus of great importance. The authors included some references that studied rule-breaking in the context of COVID-19 (Carvalho et al., Dong et al., Nofal et al., O’Connell et al., Oosterhoff et al.). However, the authors do not make an explicit link between rule-breaking and adherence to preventive measures. I think the authors can increase the impact of their paper when they make this link more explicitly in the text (at least when discussing the general relevance in the discussion section), and maybe also link their results to other studies that analyzed individual differences in adherence/non-adherence to COVID-19 preventive measures (e.g., Hartmann, M., & Müller, P. (2022). Acceptance and Adherence to COVID-19 Preventive Measures are Shaped Predominantly by Conspiracy Beliefs, Mistrust in Science and Fear–A Comparison of More than 20 Psychological Variables. Psychological reports, 00332941211073656; Šrol, J., Ballová Mikušková, E., & Čavojová, V. (2021). When we are worried, what are we thinking? Anxiety, lack of control, and conspiracy beliefs amidst the COVID‐19 pandemic. Applied Cognitive Psychology, 35(3), 720–729). For example, Hartmann and Müller also reported correlations between norm compliance and various personality traits (e.g., neuroticism, extraversion, anxiety, intuitive thinking).

1.6, goal 4: the authors wrote that they will investigate the relationship between personality, behavior, and cognitive processes. It should be mentioned more precisely how it is planned to analyze this, e.g., which parameters of the mouse trajectory are planned to be correlated with which personality traits? Also in the method section this should be specified more precisely. For example, if all possible mouse trajectory parameters are correlated with about 8 possible personality traits, there is a big number of tests and the problem of false positive detection (alpha-cumulation) occurs, it should be mentioned in the method section how many tests are conducted and if/how tests are corrected for multiple comparison. As I can see from the supplemental material, around 60 correlations have been computed. The problem of alpha cumulation should at least be mentioned somewhere.

p. 16: it is a bit unusual that only the first quartile (25%) is classified as rule-breaker and compared to the rest. Are the results different when a median split is used, or when the first quartile is compared only to the fourth quartile? Maybe I misunderstood this, because on p. 16 it is written that about 49.2% of participants were classified as rule-breakers. Please clarify.

2.4 data analys -> it should be described in more details which mouse tracking parameters are computed and how this is done (e.g., outlier trajectories can have a great impact on average trajectories – how is this controlled?)

For the questionnaires, I did not understand why the correlations are computed separately for rule-followers and rule-breakers. Doesn’t it make more sense to consider this as a continuum, or respectively to try to predict rule-followers and rule-breakers by the personality traits by means of logistic regressions? Relatedly, I understand that you added the correlation table in the supplemental materials, but since these data is essential for some of your hypothesis, I would appreciate to see (some of the) correlational results also in the main document.

p. 3 “this gap is part rooted” -> partly

p. 12 technical specificities: 75Hz was the refresh rate of the monitor, was this also the tracking rate for the mouse? Which mouse was used?

p. 15 I am wondering whether 7 trials for the neutral condition are enough to create a valid average mouse trajectory.

Results. The figure with mouse trajectory of rule-breakers vs. rule-followers should be included in the main document (not supplementary material). Also, in Figure 2, please add a legend for yellow = X and pink = Y in the Figure.

6. PLOS authors have the option to publish the peer review history of their article (what does this mean?). If published, this will include your full peer review and any attached files.

Reviewer #1: **Yes: **Miguel Clemente

Reviewer #2: No

---

## [Author Response · Author response to Decision Letter 0]

1 Sep 2022

Dear Prof. Garcia-Segarra,

We sincerely thank you and the reviewers for the time and effort you invested in reviewing our manuscript. We appreciate the helpful comments and worked hard on implementing all of them in our revised version.

We addressed each specific comment as outlined in the letter we sent to you here, and highlighted in yellow single changes in the revised version of the manuscript.

---

## [Decision Letter · Decision Letter 1]

6 Sep 2022

Taking the chance! – Interindividual differences in rule-breaking

PONE-D-22-11962R1

Dear Dr. Cubillos-Pinilla,

We’re pleased to inform you that your manuscript has been judged scientifically suitable for publication and will be formally accepted for publication once it meets all outstanding technical requirements.

Kind regards,

Jaume Garcia-Segarra

Academic Editor

PLOS ONE

Additional Editor Comments (optional):

Reviewers' comments:

Reviewer's Responses to Questions

**Comments to the Author**

1. If the authors have adequately addressed your comments raised in a previous round of review and you feel that this manuscript is now acceptable for publication, you may indicate that here to bypass the “Comments to the Author” section, enter your conflict of interest statement in the “Confidential to Editor” section, and submit your "Accept" recommendation.

Reviewer #2: All comments have been addressed

2. Is the manuscript technically sound, and do the data support the conclusions?

Reviewer #2: Yes

3. Has the statistical analysis been performed appropriately and rigorously? 

Reviewer #2: Yes

4. Have the authors made all data underlying the findings in their manuscript fully available?

Reviewer #2: No

5. Is the manuscript presented in an intelligible fashion and written in standard English?

Reviewer #2: Yes

6. Review Comments to the Author

Reviewer #2: (No Response)

7. PLOS authors have the option to publish the peer review history of their article (what does this mean?). If published, this will include your full peer review and any attached files.

Reviewer #2: No

---

## [Editor Report · Acceptance letter]

20 Sep 2022

PONE-D-22-11962R1 

Taking the chance! – Interindividual differences in rule-breaking 

Dear Dr. Cubillos-Pinilla:

I'm pleased to inform you that your manuscript has been deemed suitable for publication in PLOS ONE. Congratulations! Your manuscript is now with our production department. 

Kind regards, 

on behalf of

Dr. Jaume Garcia-Segarra 

Academic Editor

PLOS ONE